# Influence of Optional Crystallization Firing on the Adhesion of Zirconia-Reinforced Lithium Silicate before and after Aging

**Murilo Rocha Rodrigues** [1], **Manassés Tercio Vieira Grangeiro** [1], **Natalia Rivoli Rossi** [1], **Nathalia de Carvalho Ramos** [1], **Rodrigo Furtado de Carvalho** [2], **Estevão Tomomitsu Kimpara** [1], **João Paulo Mendes Tribst** [3,*] and **Tarcisio José de Arruda Paes Junior** [1]

1.   Department of Dental Materials and Prosthodontics, Institute of Science and Technology, Sao Paulo State University (UNESP), Eng. Francisco José Longo Avenue, 777, São José dos Campos 12220-000, Brazil
2.   Health Applied Sciences Post Graduate Program, Department of Dentistry, Federal University of Juiz de Fora, Governador Valadares 36036-900, Brazil
3.   Department of Oral Regenerative Medicine, Academic Centre for Dentistry Amsterdam (ACTA), Universiteit van Amsterdam and Vrije Universiteit Amsterdam, 1081 LA Amsterdam, The Netherlands
*   Correspondence: j.p.mendes.tribst@acta.nl

**Abstract:** This study proposed to evaluate the influence of the crystallization firing process and the hydrothermal degradation on the bond strength between different reinforced glass-ceramics and resin cement. Material and Methods: zirconia-reinforced lithium silicate (ZLS) and lithium disilicate (LD) were divided into six groups according to aging simulation (baseline or after thermocycling) and restorative approach (ZLS without firing; ZLS with firing; LD with firing). ZLS and LD surfaces were etched with 5% hydrofluoric acid for 30 s and 20 s, respectively, and then received a layer of silane coupling agent (Monobond-N). Then, cylinders of resin cement (1 mm diameter × 2 mm height) were bonded onto their surfaces. The baseline samples were immersed in distilled water for 24 h before the microshear bond strength (μSBS) test, while half of the specimens were tested after 6000 cycles of thermocycling aging. The types of failures were analyzed through stereomicroscopic and scanning electron microscope. The failure modes were classified as adhesive, predominantly adhesive, cohesive in ceramic, or cohesive in cement. The μSBS data were analyzed by two-way ANOVA and Tukey's test. A restorative approach ($p = 0.000$) and aging ($p = 0.000$) affected the bond strength. The highest bond-strength values were observed in the ZLS without the optional crystallization firing. The most frequent failures were adhesive and predominantly adhesive. The cementation of zirconia-reinforced lithium silicate without the optional crystallization firing process leads to high bond-strength values with resin cement.

**Keywords:** glass-ceramics; bond strength; lithium silicate; zirconium dioxide

## 1. Introduction

Due to properties such as high compressive strength and abrasion, chemical stability, biocompatibility, a favorable aesthetic, adequate translucency, fluorescence, and a thermal expansion coefficient close to that of the dental structure, dental ceramics showed increasing acceptance over the years as a biomaterial for dental treatments [1,2].

In 1998, after the release of lithium disilicate (LD) ceramic (IPS e.max®, Ivoclar Vivadent Ltda, Schaan, Liechtenstein), glass-ceramics gained popularity due to improvements in microstructure and new processing methods [3]. Superior aesthetic quality is another factor that contributes to the attractiveness of vitreous ceramics [4,5]. Despite the great acceptance and wide use of LD, the evolution of dental materials has sought alternatives to this system through the development of glass-ceramics reinforced by other nanocrystals [6].

In this sense, a new material containing lithium silicate as the main crystalline phase in a vitreous matrix reinforced with zirconium dioxide crystals emerged [7]. These new ceramics provides good optical properties, are easily milled in through the computer-aided

design and computer-aided manufacturing (CAD/CAM) system, and achieve superior surface finishing since they have a large amount of vitreous matrix [8,9]. The major examples of zirconia-reinforced lithium silicate (ZLS) are VITA Suprinity (Vita Zahnfabrik, Bad Sachingen, Germany), available in a partially crystallized state requiring an additional thermal cycle in an ceramic oven, and Celtra Duo (Dentsply-Sirona, Bensheim, Germany), material that is available in the final phase of crystallization, which, according to the manufacturer, may or may not be taken to the oven before cementation [6].

The effectiveness of dental treatment with indirect restorations can be associated with an appropriate cementation procedure, which is dependent on factors such as ceramic material, surface treatment, and cementing agents [10,11]. Ceramic surface treatment is not standard for all-ceramic types. There are surface treatments carried out with acids for ceramics with a large amount of vitreous matrix (acid-sensitive ceramics) and surface treatments performed with the blasting of microparticles for ceramics with a large amount of crystalline matrix (acid-resistant ceramics) [12,13]. In this sense, a surface treatment that allows for better results of bond strength for one specific material but may not allow for the same results for another material with a distinct composition.

Before adhesive cementation, a silane coupling agent is recommended. It is a monomer composed of reactive organic radicals and hydrolyzable monovalent groups, which provides a chemical union between the inorganic phase of the ceramic and the organic phase of the cement, by siloxane bond [14,15]. The chemical bond promoted by silane is the principal mechanism of adhesion between some ceramics and resin cement. In addition, the silane increases the surface energy of the substrate improving the cement wetting, optimizing the microscopic interaction at the adhesive interface [16,17]. The effectiveness of silane may differ in different trademarks, shapes, and product-storage times due to its chemical instability [18].

Resin cement is an alternative to traditional cementing options. The introduction of resin cement decreased the professional working time with the reduction in post-operative sensitivity [19]. However, such advantages are not relevant if resin cement does not present an adequate union to the restoration.

In the oral cavity, chemical changes in the marginal interface are evident and demonstrated through discoloration over time [20]. The adhesive interface between the cement and the dental cavity is the most critical region in bonded restorations [21]. The chemical hydrolysis of ester bonds is considered the main reason for adhesion degradation over time, occurring concomitantly to the chewing loading [21,22]. In this sense, studies with aging simulation are required to achieve approximate results to those of events that occur in the oral environment [23,24].

In vitro aging methods have been considered an important factor in studies aiming to assess the long-term behavior of adhesives systems concerning dental material's surface degradation [25]. Different aging methods such as storage in water, thermocycling, pH cycling, and storage in sodium hypochlorite solution, as well as associations between them, have previously been reported to evaluate bonding efficacy of bonded restorations [21,25,26]. However, the test outcome clearly depends on the stress generated and the failure mechanism [26]. Regardless of that, the best in vitro aging method for studying dental materials is lacking [25,26].

Certainly, the thermal variations and fatigue are prone to occur in vivo rather than only in the hydrolytic degradation of the adhesive interface [21,26]. However, the thermocycling test is based on temperature changes that are able to induce repeated stresses at the adhesive interface, due to differences in the thermal expansion coefficients of the bonded materials leading to bonding failure [20–26].

Therefore, the present study aims to evaluate the influence of the crystallization firing process and the hydrothermal degradation on the bond strength between LD and ZLS glass-ceramics and resin cement. The null hypothesis tested was that the ceramic material does not influence the bond strength regardless of the firing cycle.

## 2. Materials and Methods

### 2.1. Materials and Study Design

Table 1 summarizes the materials used in the present study.

**Table 1.** The commercial name, manufacturers, and chemical composition of the materials used in this study.

| Material | Collective Name | Manufacturer | Composition | Batch Number |
|---|---|---|---|---|
| Celtra Duo | Zirconia-reinforced lithium silicate | Dentsply-Sirona, Bensheim, Germany | $SiO_2$, $P_2O_5$, $Al_2O_3$, $Li_2O$, $K_2O$, $ZrO_2$, $CeO_2$, $Na_2O$, $Tb_4O_7$, $V_2O_5$, $Pr_6O_{11}$, Cr, Cu, Fe, Mg, Mn, Si, Zn, Ti, Zr, and Al | 18029365 |
| IPS e.max® CAD | Lithium disilicate glass-ceramics | Ivoclar Vivadent, Schaan, Liechtenstein | 57%–80% $SiO_2$, 11%–19% $Li_2O$, 0%–13% $K_2O$, 0%–11% $P_2O_5$, 0%–8% $ZrO_2$, 0%–8% $ZnO$, 0%–5% $Al_2O_3$, and 0%–5% $MgO$ | U36613 |
| Multilink® N | Universal luting composite system | Ivoclar Vivadent, Schaan, Liechtenstein | Dimethacrylate and HEMA, inorganic particles include barium glass, ethereber trifluoride, and mixed spheroidal oxides | W44613 |
| Condac porcelana 5% | Low-viscosity gel containing hydrofluoric acid at 5% | FGM Produtos Odontológios, Joinville, SC, Brazil | 5% hydrofluoric acid, water, thickener, surfactant, and colorant | 290419 |
| Monobond N | Universal primer | Ivoclar Vivadent, Schaan, Liechtenstein | Alcoholic solution of silane methacrylate, phosphoric acid, methacrylate, and sulphide methacrylate | Y33681 |

### 2.2. Specimens Preparation

The ceramic blocks of zirconia-reinforced lithium silicate (Celtra Duo Dentsply-Sirona, Bensheim, Germany) and lithium disilicate IPS e.max CAD (Ivoclar Vivadent, Schaan, Liechtenstein) were sectioned with a diamond disk under water cooling (Isomet 1000, Buehler, Lake Bluff, IL, USA) in small specimens ($13 \times 15 \times 1.5$ mm$^3$).

The specimens had their surfaces polished with water sandpaper of decreasing granulations (800, 1200, 1500, and 2000), under constant water cooling with the use of automated Grinder-polisher (Ecomet, Buehler, Lake Bluff, IL, USA).

Then, the specimens were randomly distributed into 3 groups ($n = 20$) according to the recommended firing protocol: (a) ZLS without crystallization; (b) ZLS with crystallization; and (c) LD with crystallization (Figure 1). After the respective firings, the specimens were subdivided into the absence or presence of hydrothermal aging ($n = 10$). The crystallization cycles parameters are summarized in Tables 2 and 3.

After polishing, the ceramics were etched with hydrofluoric acid 5% for 20 s (IPS e.max CAD) and 30 s (Celtra Duo). The acid was removed with abundant air/water jet for 60 s. Then, specimens were submitted to an ultrasonic bath for 5 min and dried. Subsequently, the silane coupling agent Monobond-N (Ivoclar Vivadent, Schaan, Liechtenstein) was applied on the surfaces for 60 s, according to the manufacturer's recommendations.

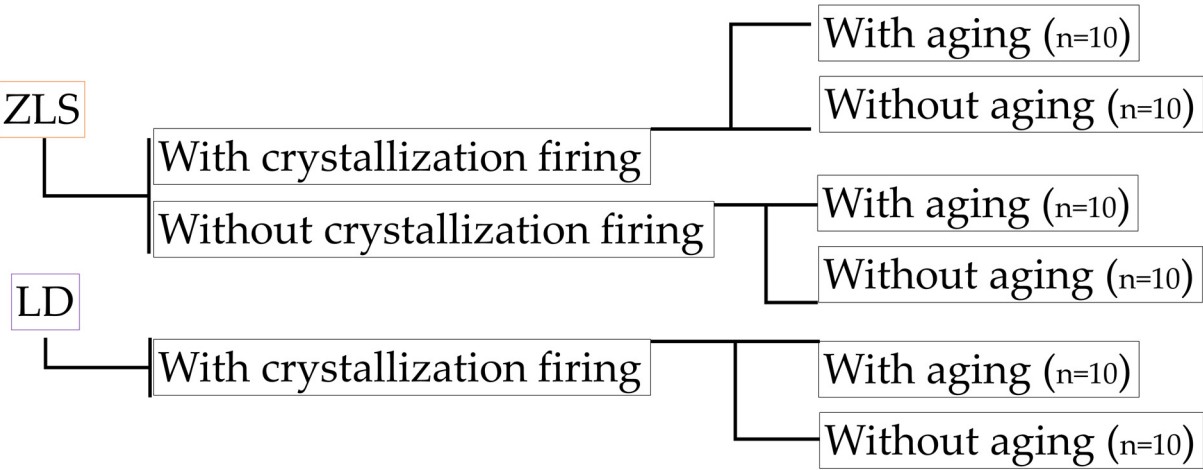

**Figure 1.** Flowchart of groups distribution according to restorative material (ZLS and LD), crystallization firing, and aging.

**Table 2.** Optional crystallization cycle according to manufacturer of ZLS.

| Firing Cycle ZLS (Celtra Duo) | |
| --- | --- |
| Initial chamber temperature | 400 °C |
| Time at initial temperature | 8 min |
| Temperature rate increase | 55 °C/min |
| Firing temperature | 830 °C |
| Holding time | 10 min |
| Ending temperature | 700 °C |

**Table 3.** Mandatory crystallization cycle according to manufacturer of LD.

| Firing Cycle LD (IPS e.max® CAD) | |
| --- | --- |
| Initial chamber temperature | 403 °C |
| Time at initial temperature | 6 min |
| Temperature rate increase | 90 °C/min |
| Firing temperature | 840 °C |
| Holding time | 7 min |
| Ending temperature | 700 °C |

*2.3. Fabrication of Resin Cement Cylinders*

The ceramic specimen was embedded into an acrylic resin cylinder with cylindrical transparent matrices (Tygon tubing, TYG-030, Saint-Gobain Performance Plastic, Miami Lakes, FL, USA) of 1 mm in an internal diameter per 2 mm in height. These matrices were fixed with wax (Wilson Polidental Ind. and Com. Ltda, São Paulo, SP, Brazil) (Figure 2).

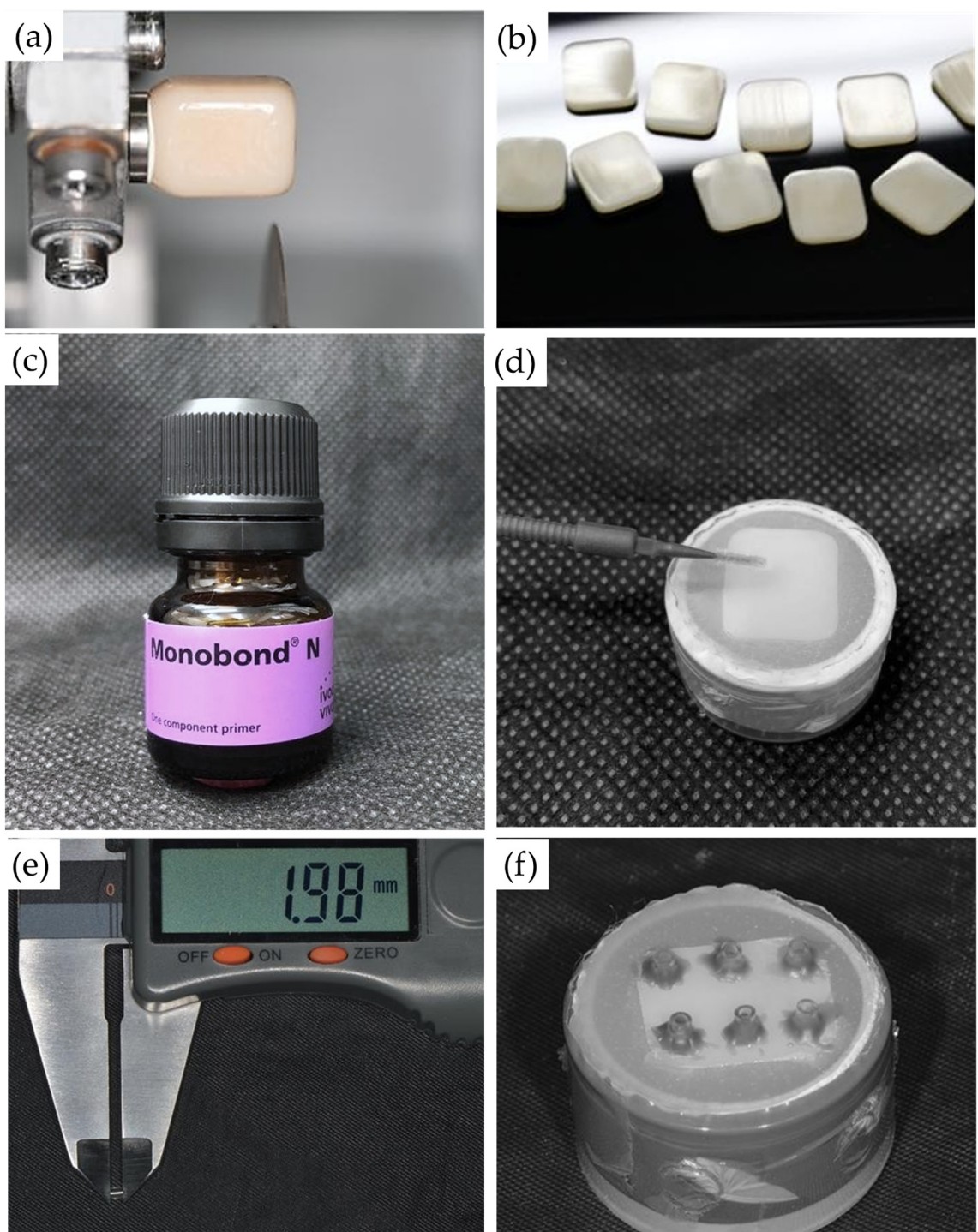

**Figure 2.** Specimen preparation. (**a**) Cutting the ceramic block at precision cutting machine, (**b**) standardized slices after cutting, (**c**) silane coupling agent used in this study, (**d**) application of silane coating on the etched ceramic surface, (**e**) cylindrical transparent matrices height, and (**f**) cylindrical transparent matrices fixed on the ceramic's surface.

The resin cement (Variolink N, Ivoclar Vivadent, Schaan, Liechtenstein) was mixed, and the material was placed in the matrices with the aid of an injection syringe (Centrix, Nova DFL, Rio de Janeiro, Brazil) and light cured for 60 s (1200 mW/cm$^2$—Radii Cal, SDI, Victoria, Australia). The matrices were removed after the light-curing, and then a resin cement cylinder was obtained on the ceramic surface (Ø = 1 mm and h = 2 mm).

After being cemented, the samples were divided according to the absence or presence of hydrothermal aging. The specimens without aging were stored in distilled water (Olidef, Ribeirão Preto, São Paulo, Brazil) at 37 °C for 24 h before the micro shear test (Figure 2).

### 2.4. Thermocycling Aging

After the preparation of the resin cylinders on the ceramic surface, half of the specimens were submitted to thermal cycling of 6000 cycles in a thermocycler (Biopdi, São Paulo, Brazil), with temperature ranging from (5 ± 1) °C to (55 ± 1) °C, with 30 s of immersion for every bath and 2 s of water flow. After thermocycling, the specimens were submitted to the micro shear test.

### 2.5. Micro Shear Bond Strength (μSBS) Test

For the μSBS test, a wire-loop design was used. A steel wire of 0.2 mm diameter was positioned at the cement/ceramic interface, and the universal test machine DL 2000 (EMIC, São José dos Pinhais, Paraná, Brazil) performed the shear test with a load cell of 50 kgf and a speed of 0.5 mm/min until failure (Figure 3).

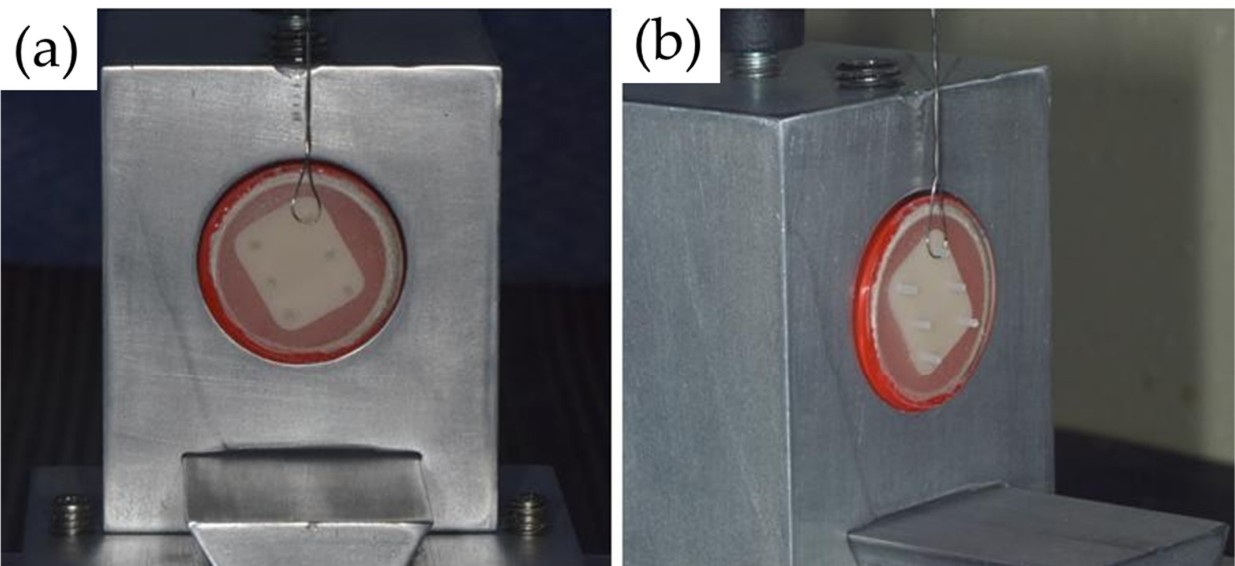

**Figure 3.** (**a**) Front view of the μSBS test; (**b**) isometric view of the μSBS test.

### 2.6. Failure Mode Analysis

After the mechanical test, a stereoscopic microscope (ZEISS MC 80 DX optical microscopy) with an increase of 50× was used to verify the fracture surfaces of the samples and determine the failure pattern at the ceramic/cement interface. Therefore, the failures were classified as adhesive (absence of cement on the ceramic surface), cohesive of the cement (cement fracture), cohesive of the ceramics (ceramic fracture), and predominantly adhesive (less than 40% cement in ceramic).

### 2.7. Scanning Electron Microscopy (SEM)

Representative samples from each experimental group were submitted to topographic analysis in scanning electron microscopy. The samples were coated with a thin layer of gold at low pressure by a sputter-coater (SC7620 Mini Sputter-Coater, Emitech, East Sussex, UK). SEM figures demonstrate the surface with and without crystallization firing.

### 2.8. Data Analysis

The μSBS data (MPa) were summarized according to descriptive statistical analysis (mean and standard deviation). For inferential statistics, the data were evaluated according

to two-way ANOVA (crystallization stage and hydrothermal aging), followed by the post hoc Tukey test with $p = 0.05$.

## 3. Results

Both factors, crystallization and thermocycling, and their interaction showed statistical differences ($p = 0.000$) (Table 4). Table 5 shows the values of the mean and standard deviation of each group.

**Table 4.** Two-way ANOVA for μSBS data.

|  | df | SS | Ms | F | *p*-Value |
|---|---|---|---|---|---|
| Thermocycling | 1 | 1444.9 | 1444.94 | 105.19 | 0.000 * |
| Firing | 2 | 1020.9 | 510.46 | 37.16 | 0.000 * |
| Thermocycling x firing | 2 | 536.5 | 268.25 | 19.53 | 0.000 * |
| Error | 353 | 4848.9 | 13.74 |  |  |
| Total | 358 | 7859.3 |  |  |  |

\* *p*-value indicates significant difference in μSBS ($p < 0.05$). Abbreviations: df degrees of freedom; SS sum of squares; and Ms: mean square.

**Table 5.** Bond strengths mean and standard deviation according to the experimental groups.

| Group | Termocycling | Bond Strengths * |
|---|---|---|
| ZLS without firing | No | 25.43 (4.06) [A] |
|  | Yes | 19.05 (3.50) [C] |
| ZLS with firing | No | 21.99 (3.85) [B] |
|  | Yes | 16.97 (3.43) [D] |
| LD | No | 18.52 (3.76) [CD] |
|  | Yes | 17.88 (3.58) [CD] |

\* Different letters indicate statistically significant differences.

Figure 4 shows the failure modes distribution. The most frequent failures were "adhesive" in all groups, followed by the "cohesive ceramic" and "predominantly adhesive" failures, respectively. After thermocycling, there was an increase in "predominantly adhesive" failures.

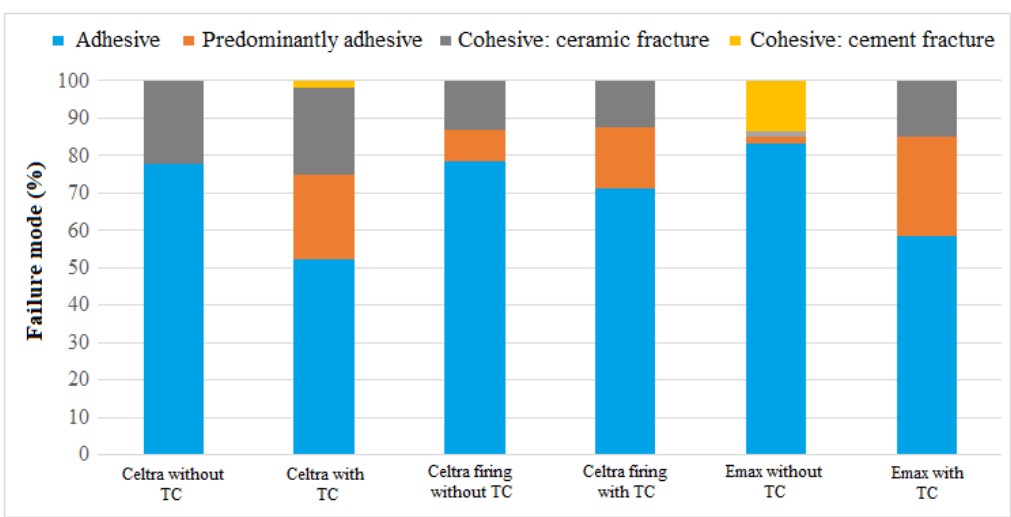

**Figure 4.** Bar graph of failure modes distribution (TC = thermocycling).

Figures 5 and 6 shows the micrographs of the groups according to the crystallization stage (without crystallization firing and with crystallization firing) and etching with HF 5%.

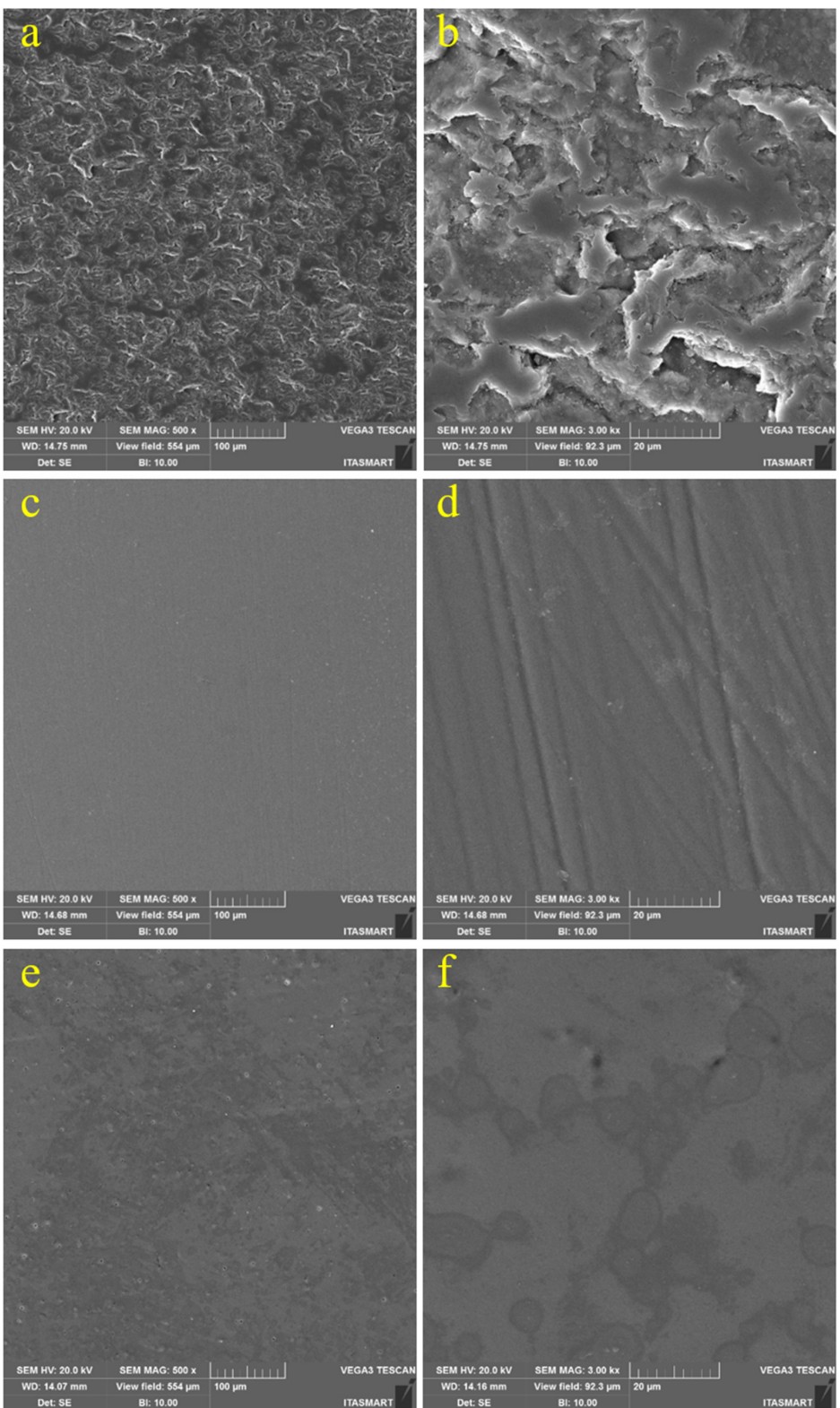

**Figure 5.** SEM images before the surfaces etching: (**a**,**b**) ZLS without crystallization firing, (**c**,**d**) ZLS with crystallization firing, and (**e**,**f**) LD with crystallization firing.

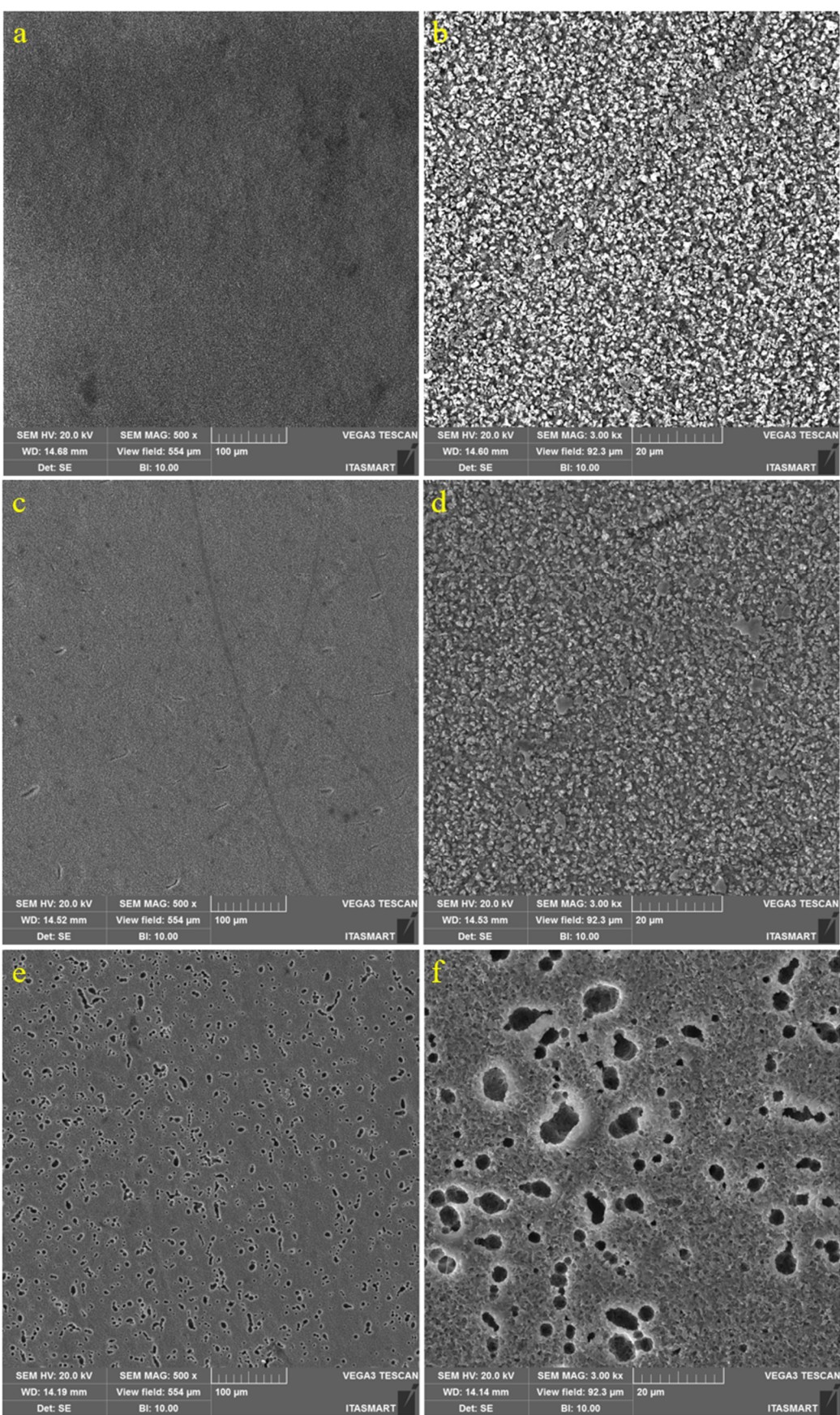

**Figure 6.** SEM images after the surfaces etching (HF 5%): (**a**,**b**) ZLS without crystallization firing, (**c**,**d**) ZLS with crystallization firing, and (**e**,**f**) LD with crystallization firing.

Figure 5 shows the images of the ceramic without acid conditioning; Figure 5a,b shows an irregular surface of the ZLS ceramic without crystallization; Figure 5c,d shows an more homogeneous surface of the ZLS ceramic when submitted to crystallization firing; and Figure 5e,f shows the LD ceramic surface (Figure 6a,b).

Figure 6 presents the images in SEM after surface etching with HF 5%, indicating the dissolution of the glassy matrix and consequently the exposure of the crystals.

## 4. Discussion

The present study evaluated the influence of the crystallization firing process and hydrothermal aging on the μSBS between the different ceramics (ZLS and LD) and the resin cement. The results showed that thermal aging reduces the bond strength between ceramics and resin cement regardless of the evaluated group, which could be justified due to the hydrolytic degradation process of the adhesive interface [6]. Therefore, the null hypotheses has been rejected.

The ceramic materials used in the present study are commercially available in blocks for milling in CAD/CAM, which are claimed to be uniform without intrinsic defects. This controlled manufacturing process could have minimized the bond-strength values, as well as the standard deviation values. Additionally, since the cemented cylinders had an area smaller than 1 mm$^2$, the mechanical test, namely, μSBS, has advantages when compared to the "macro" test with lower incorporation of defects [27].

μSBS has fewer pre-test failures results when compared to the micro-tensile bond strength test since the samples do not require processing after the cementation procedure. Therefore, residual stress and artifacts originated by cutting, which could generate lower bond-strength values, are reduced when μSBS is used [28].

The hydrothermal aging decreased the values of the bond strength. In this study, a total of 6000 cycles were used in the test machine [12]. It is known that 6000 cycles can reduce the bond strength between the resin cement and different generations of zirconia materials [29]. Therefore, the present study complemented this information showing that LD and ZLS bond strength can also be affected by this amount of thermocycling. The incorporation of crystalline nanoparticles into the glassy matrix enables both of these dental ceramics to reach high fracture resistance [30]; however, the reduction generated by the sliding contact shows a similar wear pattern shape between them, while ZLS is more resistant to wear than LD, with less volume loss and shallower surface defects [30].

For ZLS, the topographies without crystallization were more irregular before acid etching. After acid etching, regardless of the group, greater evidence of the crystals can be observed. In the groups tested, adhesive failures were the most prevalent, with a 70% rate. According to Figure 4, the ZLS ceramic was more resistant to fracture after the crystallization process due to a reduction in the percentage of cohesive ceramic failures. This fact was also found in another study, which reported the superior mechanical behavior of ceramics after the crystallization [31].

The manufacturer recommends the conditioning of the surface of ZLS (Celtra Duo) with HD between 5% and 9% for 30 s. In this study, hydrofluoric acid was used at 5%, which was the less aggressive of the recommended HF solutions. The crystals' exposure caused by the acid etching may have positively affected the bond between the resin cement and ZLS compared to LD due to a uniform distribution of crystals on the ceramic surface. The literature has reported that superior values of μSBS were found using the same protocol used in the present study [32,33].

A previous in vitro study evaluated the effect of two surface treatments (acid etching and sandblasting) on the shear bond strength of the fully-crystallized ZLS ceramic following thermocycling. The authors found that the hydrofluoric acid group showed a statistically significantly higher shear-bond-strength mean value (10.81 MPa) than aluminum oxide sandblasting (7.76 MPa) for ZLS. According to them, this behavior could be justified by the action of acid etching creating a coarse surface on the adhesive interrace by removing the glass matrix and the second crystalline phase, thus creating irregularities within the lithium

disilicate crystals [34]. Similarly, the present study applied HF as the surface treatment since it was also the recommend treatment by the manufacturer for processing this material prior to adhesive luting. However, unlike the previous investigation, this study found 21.99 MPa as the average bond strength for the same group. The differences in the values could be associated with the use of macro-specimen in the cited study as well as HF 10%.

In order to evaluate the effect of different surface treatments on the bond durability of the fully crystallized ZLS ceramic after long-term thermocycling, another investigation showed similar µTBS with traditional glass-ceramics, including lithium disilicate ceramic and leucite-based ceramic [35]. Additionally, a more durable bond strength was found for ZLS than zirconia to resist the aging process. According to the authors, the combination of HF and universal adhesive treatments is the most promising method to treat ZLS, allowing it to achieve $36.1 \pm 4.4$ MPa for before and $25.5 \pm 4.2$ MPa after aging [35]. Corroborating with the previous investigation, this study also reported a decrease in the bond strength after aging process, for both fully-crystalized and non-fired ZLS.

There is lack of data about the effect of the optional firing cycle on the ZLS bond strength. However, another study assessed the response of pre-crystallized and crystallized ZLS to diamond machining during dental milling and adjusting procedures simulation [36]. It was explained that crystallization is important to glass-ceramics since it transforms all lithium metasilicate ($Li_2SiO_3$) crystals in the glass matrix to lithium disilicate ($Li_2Si_2O_5$) crystals. The authors showed that pre-crystallized ZLS has higher removal rates with lower milling forces and lower coefficients of friction than milling ZLS in a crystallized state. However, the disadvantage of the process was extra edge chipping damage induced to the pre-crystallized material, requiring post-surface polishing to diminish the surface and subsurface damage [36]. Therefore, the present study complements these findings, suggesting that not applying the optional firing cycle may led to higher bond-strength data, but one should be cautious to ensure the proper restoration longevity.

In this study, the finishing protocol that requires additional firing was not performed, which is often necessary for the application of staining characterization (pigments) and the glaze layer. Therefore, new studies should be developed to evaluate these variables. Additionally, clinical studies should be conducted in the short, medium, and long-term to evaluate the longevity of ZLS material. Although the ZLS group without crystallization presents higher values of µSBS, the ZLS group with additional crystallization has superior mechanical resistance, which can affect its clinical application [33]. Additionally, it is worth mentioning that only one aging method was employed, without other parameters for adhesive interface degradation such as mechanical fatigue [24], pH variation, biofilm formation, or surface treatment [37]. The samples were ideally treated, and there were no contaminants on the surface. Furthermore, new studies should be developed to investigate the optical and mechanical characteristics of crystallization firing and characterization firing on ZLS.

## 5. Conclusions

Within the study limitations, it can be concluded that the cementation of zirconia-reinforced lithium silicate (Celtra Duo) without the optional crystallization firing process leads to high bond-strength values with resin cement. In addition, there was a reduction in the bond-strength values between the glass-ceramics and the resin cement after hydrothermal aging, regardless of the firing cycle or the restorative material.

**Author Contributions:** Conceptualization, M.R.R., M.T.V.G., N.R.R., N.d.C.R., R.F.d.C., E.T.K., J.P.M.T. and T.J.d.A.P.J.; methodology, M.R.R., M.T.V.G. and N.R.R.; formal analysis, M.R.R., M.T.V.G., N.R.R. and N.d.C.R.; investigation, M.R.R., M.T.V.G., N.R.R., N.d.C.R., R.F.d.C., E.T.K., J.P.M.T. and T.J.d.A.P.J.; resources, N.d.C.R., R.F.d.C., E.T.K., J.P.M.T. and T.J.d.A.P.J.; data curation, M.R.R., M.T.V.G., N.R.R. and T.J.d.A.P.J.; writing—original draft preparation, M.R.R., M.T.V.G., N.R.R. and R.F.d.C.; writing—review and editing, N.d.C.R., E.T.K., J.P.M.T. and T.J.d.A.P.J.; supervision, N.d.C.R., E.T.K., J.P.M.T. and T.J.d.A.P.J.; project administration, N.d.C.R., R.F.d.C., E.T.K., J.P.M.T. and T.J.d.A.P.J. All authors have read and agreed to the published version of the manuscript.

**Funding:** This research received no external funding.

**Institutional Review Board Statement:** Not applicable.

**Informed Consent Statement:** Not applicable.

**Data Availability Statement:** Data are available on request of the last author.

**Conflicts of Interest:** The authors declare no conflict of interest.

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
