# Peer review of "Influence of Optional Crystallization Firing on the Adhesion of Zirconia-Reinforced Lithium Silicate before and after Aging"

_coatings, doi:10.3390/coatings12121904_

Round 1

Reviewer 1 Report

1.     Please explain all non-standard abbreviation when first used in the abstract and main manuscript.

2.       What is the novelty of the present study? What are its potential clinical implictions?

3.       Since hydrothermal effects on bond strength are a major factor being evaluated in the study, specifically describe about the same in a detailed manner in the introduction and discussion.

4.       In table 1, column – composition, mention the collective name of the materials in addition to their chemical formulae

5.       There are no figures available for the methodological steps, like the blocks before and after preparation, specimens before and after polishing, etching, bonding with silane coupling agent and thermocycling.

6.       Why only half the specimens were subject thermocycling? What was the basis for the parameters simulating oral environment during thermocycling? Why two or more different thermocycling parameters were not used?

7.       In section – data analysis, enumerate all the qualitative and quantitative variables which were used.

8.       For figure 2, describing the failure modes distribution, it would be a suggestion to use a pie chart for each material (with or without thermocycling) along with the percentage for each failure mode. Also, include a legend for the abbreviation TC.

9.       In addition to the finishing protocol affecting the outcomes, are there any other limitations of the study? If yes, please mention them explicitly.

Author Response

  1. Please explain all non-standard abbreviation when first used in the abstract and main manuscript.

Dear reviewer, we have corrected the non-standard abbreviation as requested.

  1. What is the novelty of the present study? What are its potential clinical implictions?

The major novelty is the assessment of the influence of optional firing for Zirconia-reinforced lithium silicate in the bond strength. Clinically, it means that the if the dentist opt to perform the firing cycle, the professional should use a ceramic oven that costs more or less $8,500.00, and spend 30 extra minutes to finish the crown. Therefore, what are the advantages of this procedure? According to the present study, for the bond strength parameter, it is not needed.

  1. Since hydrothermal effects on bond strength are a major factor being evaluated in the study, specifically describe about the same in a detailed manner in the introduction and discussion.

The introduction has been updated as requested.

  1. In table 1, column – composition, mention the collective name of the materials in addition to their chemical formulae

The information has been added to the table 1.

  1. There are no figures available for the methodological steps, like the blocks before and after preparation, specimens before and after polishing, etching, bonding with silane coupling agent and thermocycling.

New figure with the specimen manufacturing has been added.

  1. Why only half the specimens were subject thermocycling? What was the basis for the parameters simulating oral environment during thermocycling? Why two or more different thermocycling parameters were not used?

This method is widely applied in dentistry. The immediate bond strength is measured without aging and correspond to the baseline (when the dentist finished the restoration). The other half represents the restoration after usage for a certain period; for that they are aged with thermocycling. Comparing the baseline and the aged sample’s bond strength values is possible to observe not only how protocol is the strongest but how it the most stable in long-term. The use of other thermocycling parameters is out of the scope of the present study and does not assist answering the present study’s hypothesis.

  1. In section – data analysis, enumerate all the qualitative and quantitative variables which were used.

The data are expressed in the text.

  1. For figure 2, describing the failure modes distribution, it would be a suggestion to use a pie chart for each material (with or without thermocycling) along with the percentage for each failure mode. Also, include a legend for the abbreviation TC.

The failure mode description is present on topic 2.6. The figure legends has been corrected as requested, but we opted to keep a bar chart instead pie chart.

  1. In addition to the finishing protocol affecting the outcomes, are there any other limitations of the study? If yes, please mention them explicitly.

The study’s limitations have been improved at the end of discussion section.

Reviewer 2 Report

Dear Authors,

generally, the write-up of the paper is good. 

Here are some suggestions to improve the manuscript.

Lines 67-70:

please rephrase in:

“Before adhesive cementation, a silane coupling agent is recommended. It is a monomer composed of reactive organic radicals and hydrolyzable monovalent groups, which provides a chemical union between the inorganic phase of ceramic and the organic phase of the cement by siloxane bond.”

Lines 81-82:

The authors could also outline the need for either aging simulation or cyclic fatigue.

The authors could therefore modify the following sentence, also adding two references for aging simulation and cyclic fatigue:

“In this sense, studies with aging simulation [https://doi.org/10.2341/20-156-l] and cyclic fatigue [https://doi.org/10.1111/jerd.12837 ] are required to achieve approximate results to those of events that occur in the oral environment. ”.

Both proposed references are related to lithium (di)silicate.

The present study does not include cyclic fatigue testing but is not mandatory and can be outlined in the study's limitations. (see below)

Line 84:

The authors wrote:

”Based on the exposed, the present study aimed”

please fix the sentence

Introduction and Discussion.

The article could benefit from using (at least) one null hypothesis that could be accepted or rejected at the beginning of the Discussion.

e.g.: hydrothermal aging does not affect bond strength values…

Lines 263-271:

Please add the lack of a cyclic fatigue test among the limitation of the study and possible new studies. The authors could also outline that restoration design may have an important role in interface degradation (as a finding of the already suggested paper:  [https://doi.org/10.1111/jerd.12837 ])

Author Response

Dear Authors,

generally, the write-up of the paper is good. 

Here are some suggestions to improve the manuscript.

Dear reviewer, thank you for your time and effort to improve the present study.

Lines 67-70:

please rephrase in:

“Before adhesive cementation, a silane coupling agent is recommended. It is a monomer composed of reactive organic radicals and hydrolyzable monovalent groups, which provides a chemical union between the inorganic phase of ceramic and the organic phase of the cement by siloxane bond.”

The sentence has been rephrased as requested.

Lines 81-82:

The authors could also outline the need for either aging simulation or cyclic fatigue.

The authors could therefore modify the following sentence, also adding two references for aging simulation and cyclic fatigue:

“In this sense, studies with aging simulation [https://doi.org/10.2341/20-156-l] and cyclic fatigue [https://doi.org/10.1111/jerd.12837 ] are required to achieve approximate results to those of events that occur in the oral environment. ”.

Both proposed references are related to lithium (di)silicate.

The present study does not include cyclic fatigue testing but is not mandatory and can be outlined in the study's limitations. (see below)

The references have been added in the introduction and the study’s limitations improved.

  Line 84:

The authors wrote:

”Based on the exposed, the present study aimed”

please fix the sentence

Thank you for this observation. The sentence has been corrected.

Introduction and Discussion.

The article could benefit from using (at least) one null hypothesis that could be accepted or rejected at the beginning of the Discussion.

e.g.: hydrothermal aging does not affect bond strength values…

A null-hypothesis has been added and rejected at the discussion section.

Lines 263-271:

Please add the lack of a cyclic fatigue test among the limitation of the study and possible new studies. The authors could also outline that restoration design may have an important role in interface degradation (as a finding of the already suggested paper:  [https://doi.org/10.1111/jerd.12837 ])

The discussion section was improved as requested.